# Validity and reliability of Indonesian version of the digital screen exposure questionnaire (DSEQ) for young children

Nur Faidar Khusnul Khatimah[1], Jusriani Jusriani[2], Desi Wulandari[3], Hirotaka Gima[1]*

1 Department of Physical Therapy, Graduate School of Human Health Science, Tokyo Metropolitan University, Arakawa, Tokyo, Japan, 2 Keanna Center, Optimum Child Growth and Development Center, Makassar, South Sulawesi, Indonesia, 3 OrthoTrimedical Care Center, Orthotic-Prosthetic and Child Development Therapy Center, Makassar, South Sulawesi, Indonesia

* gima@tmu.ac.jp

## Abstract

Studies have shown that excessive screen time in early childhood can negatively affect development. Therefore, assessing screen exposure in young children is important for preventing these negative effects. However, only a limited number of validated and reliable tools have been adapted into the Indonesian language. This study aimed to translate, culturally adapt, and evaluate the validity and reliability of the Digital Screen Exposure Questionnaire (DSEQ) for Indonesian children aged 2–5 years. This cross-sectional study included 171 caregivers. The translation and adaptation processes followed internationally accepted guidelines for patient-reported outcomes measures, with content and face validity evaluated through expert reviews. Reliability was assessed based on the COSMIN guidelines and previous studies, with internal consistency measured using Cronbach's alpha and test-retest reliability assessed in a subsample of 31 caregivers using the intraclass correlation coefficient (ICC). This study is the first cultural adaptation of the DSEQ in Indonesia and demonstrates good face and content validity, as confirmed by expert evaluations. Internal consistency (Cronbach's alpha) was strong for screen-time exposure and home media environment (0.704), media-related behaviors (0.863), and physical activity (0.768), whereas test–retest reliability across the three domains was moderate to high (ICC values: 0.514–0.946), with lower ICC values observed for the item related to parental supervision while watching television, which may vary owing to differences in household routines and parental availability. These findings support the Indonesian DSEQ as a valid and reliable tool for evaluating exposure to digital screens among children.

**Data availability statement:** All data files are available from the figshare database (https://doi.org/10.6084/m9.figshare.31337839). All relevant data are within the manuscript and Supporting Information files.

**Funding:** This research was funded by the Tokyo Global Partner Scholarship Program. The funders had no role in study design, data collection and analysis, decision to publish, or preparation of the manuscript.

**Competing interests:** The authors have declared that no competing interests exist.

## Introduction

Digital devices have become integral to children's daily lives worldwide [1]. Children as young as one year old use mobile phones, tablets, and other portable devices, and exposure to digital media is high among children aged 1–60 months [2]. The COVID-19 pandemic has contributed to increased screen time of children. A large multinational cohort study conducted across 12 countries found that besides older children engaged in online schooling, preschoolers without online learning requirements experienced a significant increase in screen time during the first lockdown compared to the pre-pandemic period [3]. Similarly, preschoolers in Indonesia spent approximately 2 h/day on screens pre-pandemic, which increased to 3 h during the pandemic [4].

According to the World Health Organization (WHO), screen exposure should be strictly limited during early childhood. The WHO guidelines recommend no screen time for children aged one year, < 1 h/day for children aged 2–4 years, and < 2 h/day of recreational screen time for children aged 5–17 years [5,6]. However, most children aged <5 years do not meet these recommendations [1]. In China, the average daily screen time among young children was approximately 2 h at age three, 4 h at age four, and 3 h at age five [7]. In Japan, a cohort study across 15 regional centers showed that 26.0%, 28.3%, and 30.0% of children aged 1, 2, and 3 years, respectively, watched television or DVDs for ≥2 h/day [8]. In India, young children had an average daily screen exposure (DSE) of 1.9 h [9]. Even among urban, low-income, and minority communities, children often begin using mobile devices at very early ages [10].

Although digital media can support children's learning and development, excessive use may lead to negative outcomes [11]. For instance, tablets have been shown to support early literacy skills, such as letter recognition [12]. In contrastingly, greater exposure to smart devices has been associated with poorer language abilities although it may improve fine motor skills [13]. In the United States, excessive screen time has been associated with behavioral and developmental problems [14]. Similarly, in Brazil, high levels of screen exposure were associated with poorer developmental outcomes among children aged <5 years [15]. In China, screen use before bedtime has been associated with higher odds of developing developmental coordination disorder [16]. In Indonesia, parents reported varying negative consequences associated with excessive device use, including poor posture, sleep disturbances, eye strain, reduced physical activity and play, exposure to inappropriate content, addiction, and weakened parent-child interactions [17].

The most common activity performed by children aged <5 years on mobile devices was watching videos (70.8%). Additionally, 59.6% of parents allowed their children to use mobile devices while engaging in daily tasks and domestic chores. Furthermore, 91.5% of parents reported that they had not been informed by pediatricians regarding the potential negative effects of mobile device use on their children's development [2]. When children use mobile devices while their parents are occupied with daily tasks and household chores, insufficient parental supervision can lead to inappropriate content exposure and excessive screen usage. Therefore, pediatricians should inquire regarding the duration of children's screen time, provide appropriate advice, and warn parents of the risk of excessive media use [2,18].

Previous research has reported that the association between excessive screen time and developmental or behavioral problems is stronger in preschool-aged children than in older children or adolescents [14]. A systematic review of children aged 0–7 years found that interventions aimed at reducing screen exposure were most effective among those aged 3–5 years [19]. As media exposure patterns vary across age groups and countries [20], and previous studies have shown that the association is more significant among preschoolers [14,19], valid and reliable country-specific tools are essential for monitoring screen time among young children [20].

However, several studies across worldwide have relied on questionnaires that only assess weekday and weekend screen duration and the types of devices used without reporting the validity or reliability of these instruments [8,14–16]. Similarly, most Indonesian studies have not reported the validity and reliability of screen-time assessments [21,22]. However, one study used a screen-time questionnaire that included validation procedures, but did not report its reliability [17]. One review emphasized the urgent need to assess the prevalence of digital screen exposure among children aged <5 years in developing countries [23].

The researchers developed a Digital Screen Exposure Questionnaire (DSEQ) for children aged 2–5 years to assess multiple aspects of young children's screen exposure across five domains, including sociodemographic characteristics, screen time exposure and the home media environment, physical activity, media-related behaviors, and parental media literacy, which showed good validity and reliability and was suitable for use in low- and middle-income countries. This instrument is expected to support researchers in assessing digital screen exposure in early childhood and considering appropriate interventions to reduce screen exposure, particularly in low- and middle-income countries. It may also contribute to the development of age-specific policies for managing digital screen exposure among preschool-aged children. Moreover, they recommended cross-cultural adaptation and validation of the DSEQ across diverse populations [20]. Considering that Indonesia is a developing country and there are limited validated tools for assessing screen exposure among young children, this study aimed to translate, culturally adapt, and evaluate the reliability and validity of the DSEQ for Indonesian children aged 2–5 years. We hypothesized that the Indonesian DSEQ would have good validity and reliability.

## Methods

### Participants

Participants were recruited using convenience sampling from multiple sources, including five kindergartens and two pediatric clinics in Makassar City (South Sulawesi, Indonesia), as well as one kindergarten in Jakarta, Indonesia. To increase the diversity of participants from different regions across Indonesia, a snowball sampling method was also used. The recruitment period for this study began on 7 February 2025 and ended on 10 July 2025. The inclusion criteria consisted of parents or caregivers of children aged 2–5 years who owned at least one television, smartphone, or other digital media device at home and provided informed consent to participate. This age range was selected because the developers of the DSEQ designed the instrument specifically for children aged 2–5 years [20]. In addition, previous studies have reported that the association between excessive screen time and developmental or behavioral problems is stronger in preschool-aged children than in older children or adolescents [14], and that excessive screen time generally increases with age [7,8]. Moreover, exclusion criteria included children diagnosed with cerebral palsy, Down syndrome, genetic disorders, or those with a history of traumatic brain injury.

### Sample size

The minimum required sample size (n = 100) was determined based on the criteria for internal consistency outlined in the COSMIN guidelines [24]. However, a more accurate approach would be to include three to five or up to 10 participants per questionnaire item [25]. A previous study assessed internal consistency using 52 items across domains 2–4 [20]. One item from domain 3 was removed from the online questionnaire (Table 2). Therefore, an internal consistency analysis was

conducted using 51 items. Therefore, a minimum sample size of 153 participants was required. The minimum number of participants for test–retest reliability (n = 30) was determined based on a previous study [20].

## Data collection

For convenience sampling, the researcher contacted and met with headmasters or teachers in kindergartens and therapists in pediatric clinics to approach parents who were willing to complete the questionnaire. For snowball sampling, initial participants were asked to recruit or refer other parents who met the study criteria, allowing the sample to expand progressively. The questionnaire was administered online and completed by caregivers in their preferred settings. Caregivers could contact the researcher if clarification was required. As this study involved minors, consent was obtained from their parents or legal guardians. The researcher sent an explanation of the study and an informed consent form in PDF format using WhatsApp. The parents or guardians were asked to read the documents thoroughly before deciding to participate. Those who agreed to participate indicated their consent by clicking on the questionnaire link provided by the researcher and completing the survey.

## Translation and cultural adaptation of the Indonesian DSEQ

The translation and cultural adaptation of the questionnaire followed a multistep process in accordance with international guidelines. The procedure consisted of ten stages. First, during the preparation stage, approval was obtained from the original developer of the DSEQ to translate and adapt the instrument for use in Indonesia. In the forward translation stage, two bilingual translators, both native speakers of Indonesian, fluent in English from human health science department independently translated the original DSEQ from English into Indonesian. The reconciliation stage involved an independent translator who examined both translations and combined them into a single reconciled version. This was followed by back translation, conducted by a different bilingual translator, who translated the reconciled Indonesian version back into English. The back translation review stage involved a detailed comparison between the back-translated and original English followed by harmonization. During the cognitive debriefing stage, the preliminary Indonesian version was tested with participants from the target population. Then, minor revisions were made based on participants' feedback, followed by proofreading and final report [26].

## Content and face validity

Following the procedures described in the original DSEQ development study, content and face validity were assessed by pediatric professionals. The experts evaluated the questionnaire items using the same rating approach and validity framework (Table 3) as the original developer study [20]. In this study, the Indonesian DSEQ was evaluated for content and face validity by a multidisciplinary of professionals specializing in child development, including pediatrics, child psychology, physical therapy, occupational therapy, speech therapy, and early childhood education. These experts evaluated the questionnaire's content and face validity, including its structure, relevance, clarity, and overall appropriateness. The experts were provided with the study objectives and a Likert-type feedback form. Each item was rated on a 4-point scale (1 = "not at all," 2 = "slightly," 3 = "moderately," 4 = "to a great extent"), while the questionnaire length was rated on a 5-point scale (0 = "very short," 1 = "short," 2 = "adequate," 3 = "long," 4 = "very long"). Experts were also encouraged to provide qualitative comments and suggestions. [20]. The questionnaire was revised based on their suggestions and the final version of the Indonesian DSEQ was distributed to the participants.

Furthermore, some validation studies, such as balance-related instruments, have evaluated construct validity, including structural validity (EFA/CFA) and convergent validity [25]. However, consistent with previous research, structural equation modeling (SEM) was not performed because the items in the DSEQ represent observable variables that produce direct outcomes rather than latent constructs [20]. For instance, items in the DSEQ domain assess observable indicators of exposure such as daily screen time duration, screen use frequency, and device availability at home.

## Reliability

The minimum required number of participants for test-retest reliability is 30, and ICC values < 0.5, 0.6–0.75, 0.76–0.9, and > 0.9 indicate poor, moderate, good, and excellent reliabilities, respectively [27]. The original questionnaire showed test–retest reliability in 30 participants over a 10-day interval [20]. Similarly, another study that developed a questionnaire to assess leisure-time screen-based media use in children aged 6–10 years has reported test–retest reliability in 35 participants over a 2-week interval [28]. In this study, test–retest reliability was assessed with 31 participants over an interval of 10–12 days, which met the minimum sample size requirement and the interval was aligned with previous studies [20,27,28]. Furthermore, the Cronbach's alpha values were interpreted using commonly applied thresholds: excellent (>0.90), good (>0.80), acceptable (>0.70), questionable (>0.60), poor (>0.50), and unacceptable (<0.50) [29].

## Statistical analysis

The test–retest reliability was assessed using the intraclass correlation coefficient (ICC) based on a two-way mixed-effects model with consistency for single measurements (ICC 3.1), as the same parents completed the screen-time questionnaire at two time points. ICC values were calculated for three domains of the DSEQ: screen-time exposure and home media environment (Domain 2), level of physical activity (Domain 3), and media related behaviors (Domain 4). Cronbach's alpha was applied to evaluate internal consistency, including only items suitable for categorical analysis. This analysis was not performed for sociodemographic characteristics (Domain 1), as the data were obtained from a previously validated instrument, nor for parental media literacy (Domain 5), since these items were not directly related to digital screen exposure (DSE). Descriptive statistics were used to summarize participants' sociodemographic characteristics, chi-square tests to assess the association between socioeconomic status and parental screen time, and all analyses were conducted using SPSS version 29.

## Ethical considerations

This study was approved by the Ethics Committee of Tokyo Metropolitan University, Arakawa Campus (Approval No. 24086). All participants gave their consent after being informed about the aims and procedures of the study. In addition, permission was obtained from the original author or developer of the Digital Screen Exposure Questionnaire (DSEQ) to translate and culturally adapt the instrument for use in the Indonesian context.

## Inclusivity in global research

Additional information regarding the ethical, cultural, and scientific considerations specific to inclusivity in global research is included in the Supporting Information (S1 Checklist).

# Results

## Demographic characteristics

A total of 171 participants (65.5% from Makassar and 34.5% outside Makassar) were included in the study, with 31.6% of the children aged 45–56 months. The gender distribution was nearly equal (52.0% boys and 48.0% girls, and the majority (85.4%) had their mothers as the primary caregivers. More than half of the parents held a diploma or higher degree, including 69.9% of mothers and 62.6% of fathers, while most were aged 31–40 years (45% of mothers and 59.1% of fathers). In terms of children's screen exposure, 50.9% had a daily screen time of less than two hours, whereas 49.1% exceeded two hours per day (Table 1).

## Translation, cultural adaptation, content and face validity

The translation and cultural adaptation process proceeded through 10 standardized stages. Approval to translate the DSEQ into Indonesian was obtained from the original developers, Professor Nimran Kaur and Professor Madhu Gupta

**Table 1. Sociodemographic characteristics.**

| No | Characteristics | n (%) |
|---|---|---|
| | | **All participants** |
| | | **n=171** |
| 1 | Age | |
| | 23-28 months | 19 (11.1%) |
| | 28-34 months | 21 (12.3%) |
| | 34-44 months | 40 (23.4%) |
| | 45-56 months | 54 (31.6%) |
| | 57-66 months | 37 (21.6%) |
| 2 | Area of Residence | |
| | Within Makassar | 112 (65.5%) |
| | Outside Makassar | 59 (34.5%) |
| 3 | Gender | |
| | Boy | 89 (52%) |
| | Girl | 82 (48%) |
| 4 | Primary caregiver | |
| | Mother | 146 (85.4%) |
| | Other | 25 (14.6%) |
| 5 | Income | |
| | >468 USD | 81 (47.4%) |
| | 250-468 USD | 37 (21.6%) |
| | <250 USD | 53 (31%) |
| 6 | Sibling | |
| | Elder | 59 (34.5%) |
| | Younger | 51 (29.8%) |
| | Elder and younger | 16 (9.4%) |
| | Only child | 45 (26.3%) |
| 7 | Family type | |
| | Nuclear family | 98 (57.3%) |
| | Joint family/family with 3 generations | 73 (42.7%) |
| 8 | Number of family member | |
| | 1-3 people | 26 (15.2%) |
| | 4-6 people | 115 (67.3%) |
| | >6 people | 30 (17.5%) |
| 9 | Formal childcare frequency | |
| | <3 days/week | 118 (69%) |
| | >3 days/week | 53 (31%) |
| 10 | Mother education | |
| | Primary school certificate | 2 (1.2%) |
| | Middle school certificate | 2 (1.2%) |
| | High school certificate | 46 (26.9%) |
| | Intermediate or diploma | 18 (10.5%) |
| | Graduate | 71 (41.5%) |
| | Professional degree or with honors | 32 (18.7%) |
| 11 | Father education | |
| | Primary school certificate | 3 (1.8%) |
| | Middle school certificate | 7 (4.1%) |

*(Continued)*

**Table 1.** (Continued)

| No | Characteristics | n (%) All participants n=171 |
|----|-----------------|------------------------------|
|    | High school certificate | 55 (32.1%) |
|    | Intermediate or diploma | 5 (2.9%) |
|    | Graduate | 67 (39.2%) |
|    | Professional degree or with honors | 34 (19.9%) |
| 12 | Socioeconomic status | |
|    | Upper middle (II) | 100 (58.5%) |
|    | Lower middle (III) | 52 (30.4%) |
|    | Upper lower (IV) | 19 (11.1%) |
| 13 | Mother age | |
|    | 20-30 years old | 76 (44.4%) |
|    | 31-40 years old | 77 (45%) |
|    | >40 years old | 18 (10.6%) |
| 14 | Father age | |
|    | 20-30 years old | 38 (22.2%) |
|    | 31-40 years old | 101(59.1%) |
|    | >40 years old | 32 (18.7%) |
| 15 | Read frequency | |
|    | <3 times/week | 60 (35.1%) |
|    | >3 time/week | 111 (64.9%) |
| 16 | Physical activity | |
|    | <3 hours/day | 121 (70.8%) |
|    | 3 hours or more/day | 50 (29.2%) |
| 17 | Screen time | |
|    | <2 hours | 87 (50.9%) |
|    | >2 hours | 84 (49.1%) |

from India. Two bilingual translators from the Graduate School of Human Health Science, Tokyo Metropolitan University, independently conducted forward translations, which were subsequently reconciled into a single version by comparing both translations and selecting the most accurate wording while preserving the original conceptual meaning. An example of a reconciled version is when the first translator wrote "kapan anak anda lahir" and the second translator wrote "kapankah bulan dan tanggal lahir anak"; these were then combined into "tuliskan tanggal lahir anak." This reconciled version was then back-translated by an independent bilingual linguist from Hasanuddin University, Indonesia, and the back-translated version demonstrated conceptual equivalence with the original instrument. Harmonization confirmed no major inconsistencies. Cognitive debriefing was conducted with five Indonesian parents from the target population to evaluate clarity and cultural appropriateness; based on their feedback, minor modifications were made. The final version underwent proofreading to ensure grammatical accuracy and clarity, and a comprehensive report of all adaptations was documented (Table 2).

Following translation, face validity was assessed by 12 experts specializing in child development, including one pediatrician, three child psychologists, three pediatric physical therapists, three pediatric occupational therapists, one pediatric speech therapist, and one kindergarten teacher. Using a Likert-type scale (range 1–4), the majority of items received

**Table 2. Reason for modification for Indonesian DSEQ.**

| Name of the domain | Reason for Modification |
|---|---|
| I.Sociodemographic characteristics | Modifications to the Google Form Version Only<br>1.The name of the child was replaced with initials, and full addresses were changed to current residence to protect privacy and confidentiality.<br>2.Additional explanations were provided for the nuclear family, joint family, and family with three generations to prevent participant misunderstanding.<br>3.Items related to socioeconomic status based on the Kuppuswamy scale and per capita income were removed from the direct questionnaire fields due to concerns that participants may not clearly understand these items in a self-administered online format. However, socioeconomic status can still be estimated from other information such as head of household occupation, parental education, family income, and total number of family members [30].<br>Modifications to the Google Form and Paper-Based<br>4.The order of primary caregiver response options was revised to list grandmother before grandfather, as grandmothers are more often the main caregivers in Indonesia.<br>5.Gender response options were limited to boy and girl, as additional gender categories were not considered culturally suitable in the Indonesian context.<br>6.Religion response options were revised to align with the six officially recognized religions in Indonesia (Islam, Protestantism, Catholicism, Hinduism, Buddhism, and Confucianism) [31], with an extra other, please specify option.<br>7.The order of residence categories was adjusted to list urbanized village before resettlement colony, as the resettlement colony is uncommon except in post-disaster situations.<br>8.The currency for household income was converted to Indonesian Rupiah (IDR), and the income ranges were updated according to the 2025 Modified Kuppuswamy socioeconomic classification scale [30]. |
| II.Screen time exposure and home media environment | Modifications to the Paper-Based Questionnaire Only<br>9.For item 18, sub-item labels (18.1–18.6) were added to clarify the response format.<br>Modifications to the Google Form Version Only<br>10.For item 18.04, a detailed explanation was added to clarify that supervision means physically accompanying the child and actively engaging (e.g., explaining the content), rather than simply being present in the same room.<br>11.For item 19, detailed definitions were added to clarify the types of digital programs accessed by the child. Five response options were provided based on expert recommendations: YouTube, YouTube Kids, short-form video platforms, parent-curated content only, and an "other, please specify" option.<br>12.For item 25.4, clarification was added that playing with the child means not doing other activities like watching TV, using the phone, or working at the same time.<br>Modifications to the Google Form and Paper-Based<br>13.For item 21, the instruction was changed to: "If no, skip to question 23."<br>14.For item 23, the instruction was changed to: "If no, skip to question 25." |
| III.Level of physical activity | Modifications to the Google Form Version Only<br>15.The total duration of physical activity was removed as a separate item because experts noted that it caused confusion. Asking only about weekday and weekend activity time was seen as enough. |
| IV.Media related behaviors | No further modifications were made to items that were already considered clear and culturally appropriate based on content validity results and expert evaluation. |
| V.Media literacy of the parents | No further modifications were made to items that were already considered clear and culturally appropriate based on content validity results and expert evaluation. |

mean scores above 3 (Table 3). However, experts noted that the paper-based format appeared lengthy and visually complex, potentially discouraging complete responses. This finding was consistent with the results of the cognitive debriefing, in which most parents did not complete all items, especially item number 18 domain 2. To address this, the questionnaire was converted into a Google Form by Google to improve layout clarity, make sure all questions were answered using the "required response" function, maintain participant confidentiality by requesting only initials, and provide explanation about questionnaire instructions. Experts agreed that while a paper-based format (S1 File) may be suitable for interview-based

**Table 3. Face validity by experts.**

| Questions | 0 | 1 | 2 | 3 | 4 | Mean |
|---|---|---|---|---|---|---|
| | – | Not at all | Minimally | To certain extent | To large extent | |
| 1. To what extent do you think that the questionnaire will be useful for researchers dealing with children with excessive screen time? | | | | 1 | 11 | 3.9 |
| 2. To what extent do you think that the questionnaire will be useful for caregivers to report the parent perceptions/problems related to screen time? | | | | 5 | 7 | 3.6 |
| 3. To what extent do you think that the questionnaire covers most patterns of usage considered to be associated with screen time? | | | | 5 | 7 | 3.6 |
| 4. To what extent do you think that the questionnaire measures screen time comprehensively? | | | | 8 | 4 | 3.3 |
| 5. To what extent is the language of questionnaire appropriate and understandable? (considering the fact that screen time is common in rural/illiterate and urban/literate population) | | | 4 | 8 | | 2.7 |
| 6. How will you rate the length of the questionnaire? | Very short | Short | Adequate | Long | Very long | |
| | | | 2 | 9 | 1 | 2.9 |

administration, a digital format (S2 File) is preferable when self-administered by parents or caregivers, provided that respondents can still contact the researcher for inquiries when needed. In addition, because two items in Domain 1 (socio-demographic characteristics) and one item (total duration) in Domain 3 (physical activity) were removed from the online questionnaire (Table 2), the total number of items for the online version was 83, whereas the paper-based version used for interview method surveys retained all 86 items.

### Test-retest reliability

A total of 31 participants completed the same questionnaire twice with 10–12 days interval. The intraclass correlation coefficients (ICC) for test–retest reliability ranged from 0.514 to 0.946 across all items, and all correlations were statistically significant ($p \leq 0.001$). The highest reliability was observed for the item "The child uses digital devices for completing homework assignments" (ICC = 0.946), whereas the lowest reliability was found for the item "How often does an adult supervise the child while watching television?" (ICC = 0.514) (Table 4).

### Internal consistency

The overall Cronbach's alpha for the combined domains (screen-time exposure and home media environment, physical activity, and media-related behaviors) was 0.838, indicating good internal consistency. Domain-wise analysis showed Cronbach's alpha values of 0.704 for screen-time exposure and home media environment, 0.768 for physical activity, and 0.863 for media-related behaviors.

### Association between socioeconomic status and parental screen time

In the chi-square analysis, socioeconomic status was significantly associated with fathers' screen time ($\chi^2 = 10.1$, $p = 0.006$), indicating that screen time patterns differed across socioeconomic groups. By contrast, no significant association was observed between socioeconomic status and mothers' screen time ($\chi^2 = 1.78$, $p = 0.41$).

## Discussion

The Indonesian DSEQ (online version) is a valid and reliable instrument for assessing screen exposure among young Indonesian children, with strong internal consistency and moderate-to-high test-retest reliability. Previous studies have

**Table 4. Test-retest reliability.**

| Domain and Item | | ICC | 95% CI | Sig. |
|---|---|---|---|---|
| **Domain 2: Screen exposure and home media environment** | | | | |
| 1 | What is the frequency of watching television in a typical week? | 0.663 | 0.409–0.822 | <0.001 |
| 2 | Duration of watching television on a typical working day? | 0.745 | 0.535–0.868 | <0.001 |
| 3 | Duration of watching television on a typical holiday? | 0.874 | 0.755–0.937 | <0.001 |
| 4 | Does the child watch television supervision frequently by an adult? | 0.514 | 0.201–0.732 | 0.001 |
| 5 | What is the frequency of using smartphone in a typical week? | 0.618 | 0.342–0.795 | <0.001 |
| 6 | Duration of using smartphone on a typical working day? | 0.656 | 0.398–0.818 | <0.001 |
| 7 | Duration of using smartphone on a typical holiday? | 0.521 | 0.210–0.737 | 0.001 |
| 8 | Does the child use smartphone supervision frequently by an adult? | 0.539 | 0.234–0.748 | <0.001 |
| 9 | What is the frequency of using other gadgets in a typical week? | 0.541 | 0.237–0.749 | <0.001 |
| 10 | Duration of other gadgets on a typical working day? | 0.561 | 0.264–0.761 | <0.001 |
| 11 | Duration of other gadgets on a typical holiday? | 0.647 | 0.385–0.813 | <0.001 |
| 12 | Does the child use other gadgets supervision frequently by an adult? | 0.601 | 0.319–0.786 | <0.001 |
| 13 | Do you have any rules regarding when, where, what & how to watch digital screen? | 0.614 | 0.337–0.793 | <0.001 |
| 14 | Average duration of screen time per day of the mother | 0.674 | 0.424–0.828 | <0.001 |
| 15 | Average duration of screen time per day of the father | 0.802 | 0.629–0.899 | <0.001 |
| **Domain 3: Level of physical activity** | | | | |
| 16 | Average duration of outside play per day on working/school days | 0.679 | 0.433–0.831 | <0.001 |
| 17 | Average duration of outside play per day on holidays | 0.705 | 0.471–0.846 | <0.001 |
| **Domain 4: Media behaviors of the child** | | | | |
| 18 | The child uses for completing their homework assignments | 0.946 | 0.891–0.973 | <0.001 |
| 19 | The child uses video calling applications to talk to family or friends | 0.796 | 0.619–0.896 | <0.001 |
| 20 | The child uses poems rhymes ABC etc. online | 0.549 | 0.247–0.754 | <0.001 |
| 21 | The child uses math numbers of tables online | 0.526 | 0.535–0.868 | 0.001 |
| 22 | The child uses to recognize shapes sounds colors when shown online | 0.535 | 0.228–0.745 | <0.001 |
| 23 | The child learns various sciences online | 0.780 | 0.592–0.887 | <0.001 |
| 24 | The child learns to draw write online | 0.656 | 0.397–0.818 | <0.001 |
| 25 | The child plays video games | 0.546 | 0.242–0.752 | <0.001 |
| 26 | The child uses digital media gadgets to watch stories | 0.678 | 0.430–0.830 | <0.001 |
| 27 | The child to watch adult programs | 0.905 | 0.813–0.953 | <0.001 |
| 28 | The child uses to learns letters words vocabulary language online | 0.59 | 0.303–0.779 | <0.001 |
| 29 | The child uses to watch random things for enjoyment | 0.584 | 0.294–0.775 | <0.001 |

reported good face validity and strong internal consistency for screen exposure, home media environment, media-related behaviors, and physical activity, with Cronbach's alpha values of 0.82, 0.74, and 0.73, respectively, whereas the ICC ranged from 0.56–0.99. However, the previous study used an interview method [20]. Comparisons with other countries were limited because validity and reliability evidence is currently available only from the original developer's study.

Based on the pre-test conducted with the participants and evaluations provided by child development professionals, the original DSEQ layout was considered suitable only for administration through interview surveys. However, in this study, using an interview method would not be feasible for reaching a wider geographical area of Indonesia and would be less efficient concerning time and resources. Additionally, the pre-test results showed that not all parents fully completed the paper-based questionnaire. Therefore, the questionnaire was converted into an online format with several modifications (Table 2). Furthermore, online questionnaires allow for broad and rapid data collection, enabling the inclusion of more representative samples [32], and are considered appropriate for assessing screen exposure in children [17,28,33,34].

The internal consistency results in this study were within the acceptable range, as Cronbach's alpha values should be from 0.70–0.90. Lower alpha values are often associated with a limited number of items, weak inter-item relationships, or differences in item measures. Conversely, a very high alpha value may indicate that some items are repeated, asking almost the same question in different ways [35]. The alpha values found in this study are within the ideal range, showing that the Indonesian DSEQ domains are clear.

The test-retest reliability showed the stability of the instrument over time [27]. The highest ICC value was observed for the item related to the use of digital devices to complete homework assignments, which may be owing to the stability of participant responses over the specified interval, as most children aged 2–5 years do not yet use digital devices for home-work assignments. Moreover, the lowest ICC value was found for the item related to parental supervision while watching television, which may vary owing to differences in household routines and parental availability.

In addition to questionnaires, screen exposure in children can be assessed such as through diaries, electronically prompted sampling, direct observation, recording devices (fixed or portable cameras and audio recorders), screen devices for onboard logging, and remote digital trace logging. Each method has its advantages and disadvantages. However, methods with higher accuracy and lower bias often incur higher costs, and may raise privacy concerns for children and their families, making it difficult to recruit participants [32]. Therefore, using a questionnaire is considered appropri-ate because it imposes a low burden on researchers and participants. However, this still depends on the ability of the researcher and the willingness of the participants, as methods with lower bias are preferable whenever feasible.

Another finding of this study was that socioeconomic status was significantly associated with fathers' screen time but not with mothers' screen time (Table 5). Fathers with higher socioeconomic status (level II) were more likely to report screen times exceeding 3 h, suggesting that paternal screen use may vary according to socioeconomic conditions. By contrast, mothers' screen time showed relatively similar patterns across socioeconomic levels, which may be related to their comparable caregiving roles and daily routines. Although direct comparisons of fathers' and mothers' screen time by socioeconomic status remain limited, the findings vary across countries. For example, a previous European study has reported that parents from families with a lower socioeconomic status applied more consistent rules regarding gaming and watched more television near their children; these parenting practices were associated with longer screen time [36]. In addition, studies from China have indicated that children and adolescents from lower socioeconomic backgrounds are more likely to have higher levels of screen time [37], highlighting the role of the socioeconomic context in shaping screen-related behaviors within families.

Furthermore, many existing questionnaires assessing screen exposure in children are limited in scope, with sev-eral focusing solely on screen time duration and the types of devices used and do not report validity or reliability testing

Table 5. Association between socioeconomic status and parental screen time.

| No | Characteristics | n (%) | | | | P value |
|---|---|---|---|---|---|---|
| | | All participants | Socioeconomic status | | | |
| | | | II | III | IV | |
| | | n=171 | n=100 | n=52 | n=19 | |
| 1 | Mother screen time | | | | | 0.41 |
| | <3 h | 116 (67.8%) | 65 (65%) | 39 (75%) | 12 (63.2%) | |
| | >3 h | 55 (32.2%) | 35 (35%) | 13 (25%) | 7 (36.8%) | |
| 2 | Father screen time | | | | | 0.006* |
| | <3 h | 94 (55%) | 46 (46%) | 38 (73.1%) | 10 (52.6%) | |
| | >3 h | 77 (45%) | 54 (54%) | 14 (26.9%) | 9 (47.4%) | |

*Statistical significance was set at p<0.05

[8,14–16,21,22,38]. However, evaluating other dimensions is crucial to understanding the impact of digital exposure on child development. Contrastingly, the DSEQ is a comprehensive tool that assesses multiple aspects of digital screen exposure and has been shown to be valid and reliable [20]. Knowingly, this is the first study to translate and culturally adapt the DSEQ into the Indonesian language, making Indonesia the first country to evaluate this instrument outside of its country of origin.

Despite its strengths, this study had several limitations. First, the sample was predominantly drawn from Makassar City, which may limit the generalizability of the findings. Second, the data were collected using convenience and snowball sampling, which may have introduced sampling bias. Third, the screen time data were based on parental reports (questionnaire) and may have been subject to recall or reporting bias. However, convenience and snowball sampling and questionnaire-based measures are widely used and considered acceptable in similar contexts, and have been applied in several previous studies on screen time [39–42]. Moreover, convenience and snowball sampling were employed to facilitate participant access, increase the sample size, and enhance participant trust, particularly because the survey was administered online and link-based scams have become common in Indonesia.

## Conclusions

The Indonesian version of the DSEQ showed good face and content validity as confirmed by experts, acceptable to excellent internal consistency across domains, and moderate-to-high test–retest reliability. Thus, the Indonesian DSEQ (online version) may be a valid and reliable tool for assessing digital screen exposure in early childhood and may be useful for research, clinical practice, and public health programs in Indonesia. Future studies are recommended to evaluate the Indonesian DSEQ in larger and more diverse populations across Indonesia to enhance its generalizability and to evaluate the validity and reliability of paper-based questionnaires using the interview method. Moreover, the Indonesian DSEQ could be used in future research to investigate the association between digital screen exposure and various aspects of child development.

## Supporting information

**S1 File. Paper-based digital screen exposure questionnaire (DSEQ).**
(DOCX)

**S2 File. Google-form digital screen exposure questionnaire (DSEQ).**
(DOCX)

**S1 Checklist. Inclusivity in global research.**
(PDF)

## Acknowledgments

We are grateful for the contributions of Raden Galuh Gurmadi and Lavi Aida Masniar as forward translators, and Dra. Herawaty, M.Hum., M.A., Ph.D., as back translator. We also thank the professionals specializing in child development for their suggestions, the kindergarten headmaster and teachers for giving permission to collect data in their schools, colleagues who helped share the questionnaire, and especially the parents or caregivers who participated in this study.

## Author contributions

**Conceptualization:** Nur Faidar Khusnul Khatimah, Hirotaka Gima.

**Data curation:** Nur Faidar Khusnul Khatimah, Hirotaka Gima.

**Formal analysis:** Nur Faidar Khusnul Khatimah, Hirotaka Gima.

**Funding acquisition:** Nur Faidar Khusnul Khatimah, Hirotaka Gima.

**Investigation:** Nur Faidar Khusnul Khatimah, Jusriani Jusriani, Desi Wulandari.

**Methodology:** Nur Faidar Khusnul Khatimah, Hirotaka Gima.

**Project administration:** Nur Faidar Khusnul Khatimah, Hirotaka Gima.

**Resources:** Nur Faidar Khusnul Khatimah, Hirotaka Gima.

**Software:** Nur Faidar Khusnul Khatimah, Hirotaka Gima.

**Supervision:** Hirotaka Gima.

**Validation:** Nur Faidar Khusnul Khatimah, Hirotaka Gima.

**Visualization:** Nur Faidar Khusnul Khatimah, Hirotaka Gima.

**Writing – original draft:** Nur Faidar Khusnul Khatimah.

**Writing – review & editing:** Nur Faidar Khusnul Khatimah, Hirotaka Gima.

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
