## [Decision Letter · Decision Letter 0]

28 Dec 2025

Dear Dr. Gima,

Thank you for submitting your manuscript to PLOS ONE. After careful consideration, we feel that it has merit but does not fully meet PLOS ONE’s publication criteria as it currently stands. Therefore, we invite you to submit a revised version of the manuscript that addresses the points raised during the review process.

We look forward to receiving your revised manuscript.

Kind regards,

Yagnik Dave

Academic Editor

PLOS One

Journal Requirements:

5. Please amend your authorship list in your manuscript file to include author Jusriani Jusriani..

6. Please amend the manuscript submission data (via Edit Submission) to include author Jusriani.

Additional Editor Comments

Dear Authors,

A major revision is required to meet the scientific needs of the journal.

Reviewers' comments:

Reviewer's Responses to Questions

**Comments to the Author**

1. Is the manuscript technically sound, and do the data support the conclusions?

Reviewer #1: Yes

Reviewer #2: Partly

2. Has the statistical analysis been performed appropriately and rigorously?

Reviewer #1: Yes

Reviewer #2: No

3. Have the authors made all data underlying the findings in their manuscript fully available?

Reviewer #1: Yes

Reviewer #2: No

4. Is the manuscript presented in an intelligible fashion and written in standard English?

Reviewer #1: Yes

Reviewer #2: Yes

Reviewer #1: -The abstract can briefly summarize which items in the translated scale have been changed. -The test-retest sample size (n=31) must be clearly stated in the abstract (the authors have mentioned it, but it could be made more visible).

-The main message could be more impactful: “This study is the first cultural adaptation of the DSEQ in Indonesia.”

-More could be said about the effects of parental guidance, content type, and media literacy on development. -The research hypothesis is not clearly stated at the end of the introduction. Suggestion:

“It is hypothesized that the Indonesian DSEQ will demonstrate good validity and reliability.”

-A minimum sample size calculation has been made in accordance with the COSMIN guidelines. However, it would be more accurate to include 3-5 or 10 people for each question in the survey. You should write the power analysis accordingly. You should also reference the following article.

References 1: Yetiş, A., Canli, M., Yildiz, N. T., Kocaman, H., Yildirim, H., Kuzu, Ş., ... & Duran, S. (2025). Investigation of the validity, reliability and psychometric properties of the Turkish version of the Ottawa sitting scale in patients with Parkinson's disease. Scientific Reports, 15(1), 2628.

Convenience + snowball sampling limits representativeness. This limitation should be more strongly addressed in the discussion.

The rationale for inclusion criteria should be explained (e.g., why only 2–5 years?).

Variables such as parental education level and socioeconomic status could be reported more systematically.

-The “acceptable to excellent” statements for Cronbach's alpha should be defined by classification. -The explanation for Domain 3 is confusing; it could be more clearly stated which items were excluded. -ICC reporting should be more standardized: the ICC model (ICC 2.1 or 3.1?) should be specified. -The confidence interval (CI 95%) should be provided. -The normality of the data or the validation of the test-retest interval should be explained.

-The discussion could be more structured:

1. Main findings

2. Literature comparison

3. Methodological advantages

4. Limitations

5. Conclusion and recommendations

-There are too many operational details; the discussion should focus on scientific interpretations.

-The relationship between socioeconomic status and parental screen-time could be addressed.

Reviewer #2: Abstract

The abstract is generally clear and acceptable. Minor refinement is suggested regarding the reporting of test–retest reliability. It would be helpful to clarify whether the reported ICC values apply consistently across all questionnaire domains and to briefly contextualise domains with lower ICC values.

Introduction

The introduction provides a comprehensive overview of screen time issues in early childhood. However, it would benefit from a clearer description of the questionnaire itself. Specifically, the authors should outline the domains covered by the DSEQ, summarise its previously reported validity and reliability, and briefly explain how the instrument contributes to understanding or mitigating excessive screen exposure. This would better justify the choice of the DSEQ and strengthen the study rationale.

Methodology

Validity assessment

The statistical methods used to evaluate validity are not described in sufficient detail. Content validity typically involves expert evaluation, whereas face validity is usually assessed by non-experts or target respondents. The manuscript should clearly explain:

Who was involved in content and face validity assessments

How ratings were performed

The source or framework used for validity evaluation

Sample size

The sample size calculation is not adequately described, and no reference is provided. As sample size for validation studies commonly depends on the number of questionnaire items, the rationale and source for the chosen sample size should be explicitly stated and cited.

Test–retest reliability

The choice of a 12-day interval for test–retest reliability is not explained. A brief justification, supported by literature where possible, would improve methodological transparency.

Questionnaire distribution

The method of questionnaire distribution to caregivers is not described. Details such as online versus paper-based administration, setting, and supervision should be provided.

Additional validity testing

It is unclear whether other forms of validity, such as construct validity or criterion validity, were assessed among participants. If not conducted, this should be acknowledged and justified.

Discussion

The discussion would benefit from deeper analytical interpretation of the findings. In particular, the rationale for selecting the 2–5-year age group should be clarified, especially as the questionnaire includes items related to homework, which may not be applicable to younger children. The limitations section should also be expanded to address methodological constraints, generalisability, and areas for future research.

**Do you want your identity to be public for this peer review?** For information about this choice, including consent withdrawal, please see our For information about this choice, including consent withdrawal, please see our Privacy Policy .

Reviewer #1: No

Reviewer #2: No

---

## [Author Response · Author response to Decision Letter 1]

13 Feb 2026

Thank you for your insightful comments and valuable suggestions, which have significantly improved our manuscript. We have addressed all comments below and explained the revisions made.

The explanation (page, number, and line) was based on Revised Manuscript with Track Changes file.

A. From Editor

Response: We have carefully checked the manuscript and ensured that it meets all PLOS ONE style and formatting requirements.

Response: We have completed the PLOS questionnaire on inclusivity in global research and included it as Supporting Information (S1 Checklist). In addition, a related subsection has been added to the Methods section (page 10, lines 221–224).

“Additional information regarding the ethical, cultural, and scientific considerations specific to inclusivity in global research is included in the Supporting Information (S1 Checklist).”

Response: We have included the complete ethics statement in the Methods section (page 10, lines 215–220).

“This study was approved by the Ethics Committee of Tokyo Metropolitan University, Arakawa Campus (Approval No. 24086). All participants gave their consent after being informed about the aims and procedures of the study. In addition, permission was obtained from the original author or developer of the Digital Screen Exposure Questionnaire (DSEQ) to translate and culturally adapt the instrument for use in the Indonesian context.”

4. When completing the data availability statement of the submission form, you indicated that you would make your data available on acceptance. We strongly recommend all authors decide on a data sharing plan before acceptance, as the process can be lengthy and hold up publication timelines. Please note that, though access restrictions are acceptable now, your entire data will need to be made freely accessible if your manuscript is accepted for publication. This policy applies to all data except where public deposits would breach compliance with the protocol approved by your research ethics board. If you are unable to adhere to our open data policy, please kindly revise your statement to explain your reasoning and we will seek the editor's input on an exemption. Please be assured that once you have provided your new statement, the assessment of your exemption will not hold up the peer review process.

Response: We have revised the data availability statement to clarify our data-sharing plan in accordance with PLOS ONE’s open data policy.

5. Please amend your authorship list in your manuscript file to include author Jusriani Jusriani.

Response: The author’s name has been corrected and updated to “Jusriani Jusriani” in the manuscript file (page 1 line 3).

6. Please amend the manuscript submission data (via Edit Submission) to include author Jusriani.

Response: The manuscript submission data have been updated to include Jusriani Jusriani as an author.

Response: We have reviewed and cited the article recommended by Reviewer 1 (Yetiş et al., 2025) in the revised manuscript.

B. From Reviewer 1

1. The abstract can briefly summarize which items in the translated scale have been changed. The test-retest sample size (n=31) must be clearly stated in the abstract (the authors have mentioned it, but it could be made more visible).

Response: We have clearly stated the test–retest sample size (n = 31) in the abstract (page 2, line 33).

“Reliability was assessed based on the COSMIN guidelines and previous studies, with internal consistency measured using Cronbach’s alpha and test–retest reliability assessed in a subsample of 31 caregivers using the intraclass correlation coefficient (ICC).”

2. The main message could be more impactful: “This study is the first cultural adaptation of the DSEQ in Indonesia.”

Response: We have added the statement highlighting that this study is the first cultural adaptation of the DSEQ in Indonesia to the abstract (page 2, line 34).

“This study is the first cultural adaptation of the DSEQ in Indonesia and demonstrates good face and content validity, as confirmed by expert evaluations.”

3. More could be said about the effects of parental guidance, content type, and media literacy on development.

Response: Additional discussion on parental guidance, media content, and media literacy has been added to the Introduction (page 4, lines 76–84).

“The most common activity performed by children aged <5 years on mobile devices was watching videos (70.8%). Additionally, 59.6% of the parents allowed their children to use mobile devices while engaging in daily tasks and domestic chores. Furthermore, 91.5% of parents reported that they had not been informed by pediatricians regarding the potential negative effects of mobile device use on their children’s development [2]. When children use mobile devices while their parents are occupied with daily tasks and household chores, insufficient parental supervision can lead to inappropriate content exposure and excessive screen usage. Therefore, pediatricians should inquire regarding the duration of children’s screen time, provide appropriate advice, and warn parents of the risk of excessive media use [2,18].”

4. The research hypothesis is not clearly stated at the end of the introduction. Suggestion: “It is hypothesized that the Indonesian DSEQ will demonstrate good validity and reliability.”

Response: The research hypothesis has been clearly stated at the end of the Introduction (page 5, lines 113–114).

“We hypothesized that the Indonesian DSEQ would have good validity and reliability.”

5. A minimum sample size calculation has been made in accordance with the COSMIN guidelines. However, it would be more accurate to include 3-5 or 10 people in each question in the survey. You should write the power analysis accordingly. You should also reference the following article. References 1: Yetiş, A., Canli, M., Yildiz, N. T., Kocaman, H., Yildirim, H., Kuzu, Ş., ... & Duran, S. (2025). Investigation of the validity, reliability and psychometric properties of the Turkish version of the Ottawa sitting scale in patients with Parkinson's disease. Scientific Reports, 15(1), 2628.

Response: We have added a new subsection describing the sample size rationale based on COSMIN guidelines and item-to-participant ratios, and cited the recommended article (Yetiş et al., 2025) in the Methods section (page 6–7, lines 131–139).

“However, a more accurate approach would be to include three to five or up to 10 participants per questionnaire item [25]. A previous study assessed internal consistency using 52 items across domains 2–4 [20]. One item from domain 3 was removed from the online questionnaire (Table 2). Therefore, an internal consistency analysis was conducted using 51 items. Accordingly, the minimum required sample size was 153.”

6. Convenience + snowball sampling limits representativeness. This limitation should be more strongly addressed in the discussion.

Response: We have expanded the discussion of sampling limitations and explained the rationale for using convenience and snowball sampling in the Discussion section (page 25–26, lines 373–380).

“Second, the data were collected using convenience and snowball sampling, which may have introduced sampling bias. Third, the screen time data were based on parental reports (questionnaire-based) and may have been subject to recall or reporting bias. However, convenience and snowball sampling and questionnaire-based measures are widely used and considered acceptable in similar contexts, and have been applied in several previous studies on screen time [39,40,41,42]. Moreover, convenience and snowball sampling were employed to facilitate participant access, increase the sample size, and enhance participant trust, particularly because the survey was administered online and link-based scams have become common in Indonesia.”

7. The rationale for inclusion criteria should be explained (e.g., why only 2–5 years?)

Response: The rationale for selecting children aged 2–5 years has been clarified in the Participants subsection of the Methods section (page 6, lines 124–128).

“This age range was selected because the developers of the DSEQ designed the instrument specifically for children aged 2–5 years [20]. In addition, previous studies have reported that the association between excessive screen time and developmental or behavioral problems is stronger in preschool-aged children than in older children or adolescents [14], and that excessive screen time generally increases with age [7,8].”

8. Variables such as parental education level and socioeconomic status could be reported more systematically.

Response: The parental education level has been reported in greater detail, and additional rows describing socioeconomic status have been added to the Results section (pages 12 and 13).

9. The “acceptable to excellent” statements for Cronbach's alpha should be defined by classification.

Response: A new Reliability subsection has been added to the Methods section, including the classification criteria for Cronbach’s alpha values (page 9, lines 198–200).

“Furthermore, the Cronbach’s alpha values were interpreted using commonly applied thresholds: excellent (>0.90), good (>0.80), acceptable (>0.70), questionable (>0.60), poor (>0.50), and unacceptable (<0.50) [29].”

10. The explanation for Domain 3 is confusing; it could be more clearly stated which items were excluded.

Response: We have clarified which item was excluded from Domain 3 in the Results section (page 18, line 269-270). The exclusion criterion was total duration of physical activity.

“In addition, because two items in Domain 1 (sociodemographic characteristics) and one item (total duration) in Domain 3 (physical activity) were removed from the online questionnaire (Table 2), the total number of items for the online version was 83, whereas the paper-based version used for interview method surveys retained all 86 items.”

11. ICC reporting should be more standardized: the ICC model (ICC 2.1 or 3.1?) should be specified.

Response: The ICC model used (ICC model specification) has been clearly reported in the Statistical Analysis subsection of the Methods section (page 9, lines 202 and 203).

The test–retest reliability was assessed using the intraclass correlation coefficient (ICC) based on a two-way mixed-effects model with consistency for single measurements (ICC 3.1).”

12. The confidence interval (CI 95%) should be provided.

Response: A new column with 95% confidence intervals has been added to the Results (pages 20–21).

13. The normality of the data or the validation of the test-retest interval should be explained.

Response: We have explained the rationale for the test–retest interval in the Methods section (page 9, lines 192–198). Previous studies have used 10-day and 14-day intervals. In our study, participants were instructed to complete the retest after 10 days; however, some participants completed it after 11–12 days, which we considered acceptable. This interval was selected to reduce recall bias while minimizing the likelihood of substantial changes in children’s daily routines.

“The original questionnaire showed test–retest reliability in 30 participants over a 10-day interval [20]. Similarly, another study that developed a questionnaire to assess leisure-time screen-based media use in children aged 6–10 years has reported test–retest reliability in 35 participants over a 2-week interval [28]. In this study, test–retest reliability was assessed with 31 participants over an interval of 10–12 days, which met the minimum sample size requirement, and the interval was aligned with previous studies [20,27,28].”

14. The discussion could be more structured: Main findings, Literature comparison, Methodological advantages, Limitations, Conclusion and recommendations

Response: The Discussion has been restructured to clearly address the main findings, comparisons with previous studies, methodological advantages, limitations, conclusions, and recommendations.

“Comparisons with other countries were limited because validity and reliability evidence is currently available only from the original developer’s study.” (page 23, lines 306 and 307).

The methodological advantages are discussed on page 24 (lines 340–348).

“In addition to questionnaires, screen exposure in children can be assessed such as through diaries, electronically prompted sampling, direct observation, recording devices (fixed or portable cameras and audio recorders), screen devices for onboard logging, and remote digital trace logging. Each method has its advantages and disadvantages. However, methods with higher accuracy and lower bias often incur higher costs, and may raise privacy concerns for children and their families, making it difficult to recruit participants [32]. Therefore, using a questionnaire is considered appropriate because it imposes a low burden on researchers and participants. However, this still depends on the ability of the researcher and the willingness of the participants, as methods with lower bias are preferable whenever feasible.”

The conclusions and future research directions are presented on page 26 (lines 382–392).

“The Indonesian version of the DSEQ showed good face and content validity as confirmed by experts, acceptable to excellent internal consistency across domains, and moderate-to-high test–retest reliability. Thus, the Indonesian DSEQ (online version) may be a valid and reliable tool for assessing digital screen exposure in early childhood and may be useful for research, clinical practice, and public health programs in Indonesia. Future studies are recommended to evaluate the Indonesian DSEQ in larger and more diverse populations across Indonesia to enhance its generalizability and to evaluate the validity and reliability of paper-based questionnaires using the interview method. Moreover, the Indonesian DSEQ could be used in future research to investigate the association between digital screen exposure and various aspects of child development.”

15. There are too many operational details; the discussion should focus on scientific interpretations.

Response: We have removed excessive operational details from the Discussion section (pages 23 and 24, lines 324–334) and moved them to the Methods section, specifically to the Reliability subsection (page 9, lines 189–200 multidisciplinary panel of professionals specializing in child development, including pediatric

---

## [Decision Letter · Decision Letter 1]

16 Mar 2026

Validity and reliability of Indonesian version of the digital screen exposure questionnaire (DSEQ) for young children

PONE-D-25-62221R1

Dear Dr.Hirotaka Gima

We’re pleased to inform you that your manuscript has been judged scientifically suitable for publication and will be formally accepted for publication once it meets all outstanding technical requirements.

Kind regards,

Yagnik Dave

Academic Editor

PLOS One

Additional Editor Comments (optional):

Reviewers' comments:

Reviewer's Responses to Questions

**Comments to the Author**

Reviewer #1: All comments have been addressed

2. Is the manuscript technically sound, and do the data support the conclusions?

Reviewer #1: Yes

3. Has the statistical analysis been performed appropriately and rigorously?

Reviewer #1: Yes

4. Have the authors made all data underlying the findings in their manuscript fully available?

Reviewer #1: Yes

5. Is the manuscript presented in an intelligible fashion and written in standard English?

Reviewer #1: Yes

Reviewer #1: (No Response)

**Do you want your identity to be public for this peer review?** For information about this choice, including consent withdrawal, please see our For information about this choice, including consent withdrawal, please see our Privacy Policy .

Reviewer #1: No

---

## [Editor Report · Acceptance letter]

PONE-D-25-62221R1

PLOS One

Dear Dr. Gima,

I'm pleased to inform you that your manuscript has been deemed suitable for publication in PLOS One. Congratulations! Your manuscript is now being handed over to our production team.

Kind regards,

on behalf of

Dr. Yagnik Dave

Academic Editor

PLOS One